# Preclinical usability evaluation of the Liveborn app: A mobile health application that provides feedback for neonatal resuscitation

Daniel Ishoso[1], Eric Mafuta[1], Kourtney Bettinger[2], Carl Bose[3], Benjamin H. Chi[4], Ingunn Haug[5], Patricia Gomez[6], Joar Eilevstjønn[5], Abigail McRea[3], Helge Myklebust[5], Antoinette Tshefu[1], Jackie K. Patterson[3]*

1 School of Public Health, University of Kinshasa, Kinshasa, Democratic Republic of Congo,
2 Department of Pediatrics, University of Kansas, Kansas City, Kansas, United States of America,
3 Department of Pediatrics, University of North Carolina at Chapel Hill, Chapel Hill, North Carolina, United States of America, 4 Department of Obstetrics and Gynecology, University of North Carolina at Chapel Hill, Chapel Hill, North Carolina, United States of America, 5 Laerdal Medical, Stavanger, Norway, 6 Jhpiego, Baltimore, Maryland, United States of America

* Jackie_patterson@med.unc.edu

## Abstract

Neonatal mortality, particularly due to failure to breathe at birth, remains a significant challenge in low- and middle-income countries (LMICs). Effective neonatal resuscitation is essential to improving survival, but is challenging to implement consistently at the bedside. The Liveborn mobile health application for newborn resuscitation was developed to provide real-time guidance and support debriefing for healthcare workers in LMICs. Liveborn allows an observer to document the timing of key actions during a resuscitation; it then compares the observer data to recommended care and provides data-driven feedback. This study aimed to evaluate the usability of Liveborn in simulated resuscitations. We conducted two rounds of simulated resuscitations using Liveborn with midwives at one health facility in the Democratic Republic of Congo. Each round included ten simulations, with half testing real-time guidance and half focusing on debriefing. Between rounds, Liveborn was iteratively refined based on analysis of video-recordings of the simulations and participant surveys. Midwives' perceptions of usability and feasibility were assessed using previously validated survey tools including the System Usability Scale (SUS) with a score >68 considered above average, and the Feasibility of Intervention Measure (FIM) with a score >12 considered above neutral. Round 1 of testing identified several key usability issues including difficulty accurately recording events, poor adherence to audio guidance that was insufficiently specific, and poor flow of debriefing for intrapartum stillbirth cases. The Liveborn app, after iterative refinement, demonstrated excellent usability (median SUS score of 90 [Q1, Q3: 85, 95]) and excellent feasibility (median FIM score of 19 [16, 20]). Further research is needed to assess Liveborn's effectiveness in real clinical settings and its impact on neonatal outcomes in LMICs.

**Data availability statement:** All data files are available from the UNC Dataverse database: https://doi.org/10.15139/S3/RSSDD4

**Funding:** JP 1R21HD103058-01 National Institute of Child Health and Human Development https://www.nichd.nih.gov/ The funder did not play any role in the study design, data collection and analysis, decision to publish, or preparation of the manuscript.

**Competing interests:** I have read the journal's policy and the authors of this manuscript have the following competing interests: JP: Received research funding from the National Institute of Child Health and Human Development, the Laerdal Foundation, the Doris Duke Charitable Foundation, the Thrasher Foundation and the Gates Foundation; she is also the recipient of a Laerdal Global Health monetary gift to support her research. CB: Received funding from the National Institutes of Health as well as travel support from the American Academy of Pediatrics and Laerdal Global Health. HM/IM/JE: Employed by Laerdal Medical, a sister company to Laerdal Global Health. BHC: Received extramural funding from the National Institutes of Health, Helmsley Charitable Trust, and UNICEF.

## Author summary

Birth asphyxia (i.e., failure to breathe at birth) is one of the top three causes of death for newborns worldwide. The vast majority of these deaths happen in low-resource settings. We developed a mobile health application called Liveborn that supports health workers in low-resource settings who are helping a newborn breathe at birth. An observer uses the app to record what is happening during a resuscitation. Using this data, the app gives audio guidance to the health worker during a resuscitation (real-time guidance) and helps them reflect on their care after a resuscitation (debriefing). We tested the usability of the app with midwives in the Democratic Republic of the Congo using a newborn manikin. In the first round of simulations, we found that it was challenging to accurately record events with the app, some of the audio guidance was not specific enough, and the screens for debriefing were not relevant for stillborn cases. We refined the app accordingly and tested it again in simulations. The refined version of Liveborn demonstrated excellent usability and feasibility. Further research is needed to assess Liveborn's effectiveness in real clinical settings and its impact on neonatal outcomes.

## Introduction

Over one million newborns die on their day of birth every year, with the majority of these deaths occurring in low- and middle-income countries (LMICs) [1]. Most of these deaths result from failure to breathe at birth (i.e., failure to breathe at birth), a treatable complication related to events occurring during labor [2]. In an effort to reduce neonatal mortality related to failure to breathe at birth, evidence-based resuscitation algorithms have been developed to improve the quality of care. Common elements in basic neonatal resuscitation algorithms include: prompt stimulation of the neonate, maintenance of euthermia, airway clearance when needed, and initiation of bag-mask ventilation (BMV) by one minute after birth [3–6]. Delivering high-quality neonatal resuscitation requires not only following the correct sequence of actions but also using appropriate technique. Adherence to basic resuscitation algorithms, including timely and continuous BMV, saves neonatal lives [7].

Simulation training is commonly implemented to teach resuscitation and provide hands-on experience in preparation for future clinical events. Helping Babies Breathe (HBB) is a neonatal resuscitation curriculum that has been well-studied in LMICs [8]. The HBB training method teaches medical providers how to efficiently and effectively perform neonatal resuscitation and has been shown to improve knowledge and skill and decrease perinatal mortality [9–12]. However, two systematic reviews have noted a decrease in knowledge and skill over time following training [13,14]. Resulting decreased adherence to recommended care can prevent training from reaching its maximal impact [15].

Feedback during or after a resuscitation event is a promising strategy to improve adherence to recommended care. For example, real-time guidance by an expert coach has been shown to increase provider preparedness, facility readiness, and quality resuscitation care [16,17]. Debriefing (i.e., discussing and analyzing clinical care) after newborn resuscitation has been shown to improve retention of knowledge and skill and enhance team effectiveness [18,19]. These feedback strategies require expert facilitators and objective data on resuscitation care, both of which can be barriers to implementation in LMICs.

To address these barriers, we developed a mobile health (mHealth) application (app) called Liveborn. This app allows an observer to document resuscitation care during the first minutes after birth and provides automated feedback for basic newborn resuscitation. The feedback

functionality supports healthcare providers with guidance *during* neonatal resuscitation and facilitates debriefing *after* resuscitation. We sought to evaluate and refine the usability of Liveborn in simulated resuscitations with midwives in the Democratic Republic of the Congo (DRC).

## Methods

### Study design and participants

We conducted a usability study to evaluate and refine the feedback functionality of Liveborn. This comprised two rounds of simulations involving midwives at a single health facility in Kinshasa, DRC with iterative refinement of the app between rounds. Midwives were eligible for inclusion if they were employed at the participating health center and routinely provided newborn care at the time of birth. The only exclusion criterion was any workplace concerns regarding participation.

### Description of the Liveborn app

Liveborn is an mHealth app to support birth attendants in low-resource settings to improve newborn resuscitation care. Using Liveborn, an observer collects granular data about events during a resuscitation such as the start and stop time of key provider actions (drying/stimulation, suctioning, BMV) and when the baby cries (Fig 1) [20]. The app connects via Bluetooth to a battery-operated heart rate meter called NeoBeat (Laerdal Global Health, Stavanger, Norway) and digitally displays heart rate when placed on the newborn's thorax [21]. Liveborn integrates data from the observer and NeoBeat to give feedback to the provider, either in real-time or following the resuscitation. While Liveborn can provide feedback relying solely on data from the observer, the objective heart rate data from NeoBeat allows for more nuanced feedback. Liveborn delivers audio-visual guidance during resuscitation to prompt the provider to follow best practices for resuscitation. Following resuscitation, Liveborn guides providers through a debriefing session by presenting data on observed care, comparing this to recommended practice, and prompting reflection with discussion questions. The feedback given by Liveborn aligns with recommendations from the International Liaison Committee on Resuscitation (ILCOR) and is specifically based on the HBB algorithm. We used a French version of Liveborn in this usability study as French is the primary language of professional healthcare workers in the DRC; the translation was verified by local clinicians to ensure accuracy and appropriateness.

### Study procedures

Each of the two evaluation rounds included ten simulations. This sample size reflects a programmatic decision based on our experience to date with the app including the types of usability problems we anticipated, the highly structured task we were testing, and the homogeneity of participants. We constructed the group of simulations to target common errors in resuscitation and a variety of app features. Five of the simulations tested Liveborn with real-time guidance, and five tested debriefing (S1 Table). For all simulations, we used NeoNatalie Live, a high-fidelity manikin designed for low-resource settings that affords the ability to set varying compliance of the lungs to support practice of increasingly complex patient scenarios [21]. NeoNatalie Live interfaces with NeoBeat to display heart rate via dry-electrode ECG during a simulation.

Prior to participation in simulations, our team oriented midwives to the simulation materials and equipment, and conducted a guided tour of the Liveborn app. Midwives had the

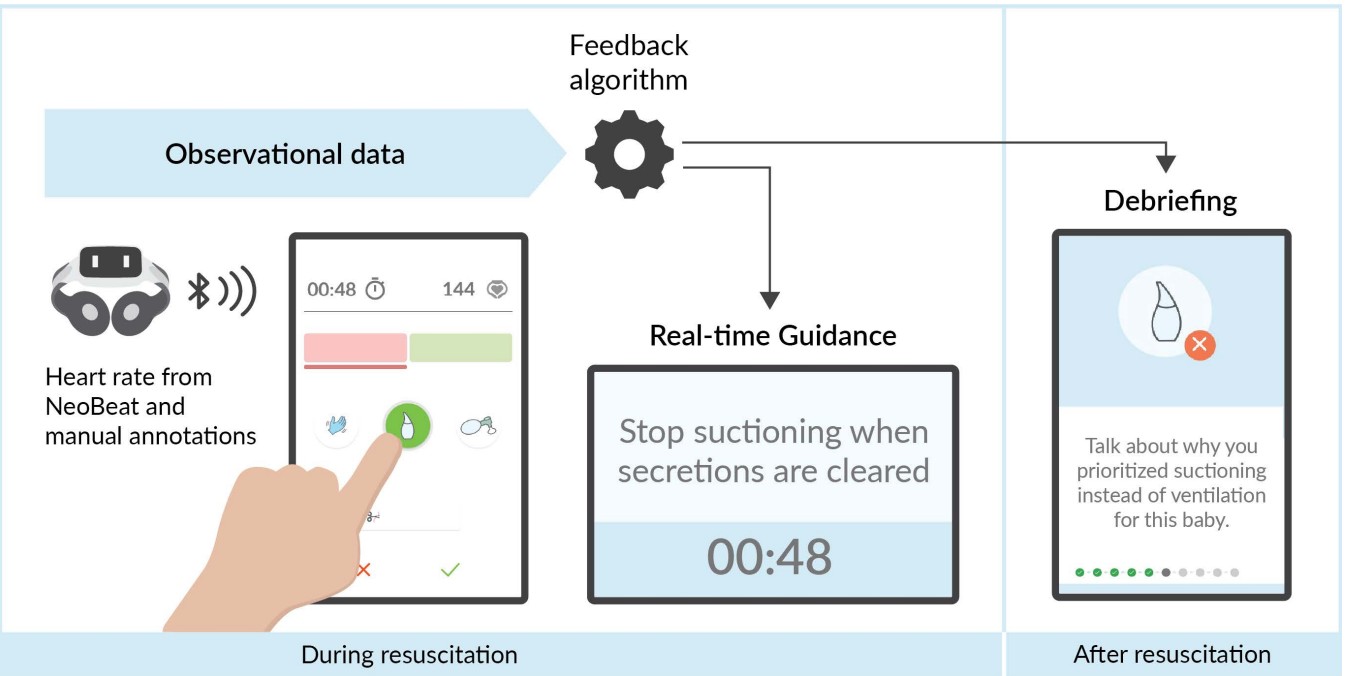

**Fig 1. Liveborn app.** Using the Liveborn app, an observer documents the timing of events during a resuscitation and the breathing status of the baby. A battery-operated heart rate meter, NeoBeat, transmits the newborn's heart rate to the Liveborn app. Using the data from the observer and NeoBeat, Liveborn provides feedback in the form of real-time guidance *during* a resuscitation and debriefing *after* a resuscitation.

opportunity to practice with all materials before the usability evaluation began. The evaluation included both event recording and feedback functionality.

We conducted Round 1 of the usability evaluation in July 2021. Midwives participated in simulated resuscitations as pairs, with one midwife acting as the newborn care provider and the other as the observer documenting the resuscitation via Liveborn. For debriefing simulations, the two midwives debriefed together. We video-recorded all simulated resuscitations and Liveborn app interactions. Following each simulation, midwives completed a survey asking about their experience with Liveborn. There were four different surveys based on the type of simulation (real-time guidance versus debriefing) and their role (provider or observer). At the end of Round 1, midwives completed two previously validated scales: the System Usability Scale (SUS) to evaluate app usability and the Feasibility of Intervention Measure (FIM) to evaluate feasibility of using Liveborn with feedback in their clinical practice [22,23].

We refined Liveborn based on the findings in Round 1, and then conducted Round 2 in October 2021 with the refined app using the same procedures described for the first round. Liveborn was further refined based on the findings from Round 2.

## Outcomes and analysis

After each round of simulations, we analyzed the video footage of midwives interacting with Liveborn to identify design problems based on an analysis plan developed a priori. Our analysis focused on the use of specific application features in each of three categories: event recording, real-time guidance, and debriefing. We triangulated this data with app usage data, post-simulation surveys, the SUS, and the FIM. Based on the findings from Round 1, we developed a summary report for the technical development team with actionable items to

improve the usability of Liveborn. Following testing of the refined Liveborn in Round 2, we completed a second analysis using the procedures outlined for Round 1.

We calculated the SUS score using standard scoring, and considered a median score above the instrument's average score of 68 to be sufficiently usable to proceed with a clinical pilot [23]. We calculated the FIM score using standard scoring, and considered a median score above neutral at >12 to be sufficiently feasible to proceed with a clinical pilot [22]. To evaluate the statistical significance of changes in scores between rounds for both the SUS and FIM, we used the Wilcoxon signed-rank test for paired samples.

### Ethics approval

All midwives participated following written informed consent. Both the University of North Carolina (UNC) and Kinshasa School of Public Health (KSPH) Institutional Review Boards (IRBs) approved this study (UNC IRB #20-3414; KSPH IRB# ESP/CE/18/2021).

## Results

### Participant demographics

Among 10 participants in this study, the median (Q1, Q3) age was 55.5 (48.3, 61.0), and each had more than 10 years of professional experience (Table 1). The majority were Registered Nurses with a Baccalaureate level of education. Half of participants owned a smartphone; among these, 60% used applications and the internet for work a few times per week (Table 1).

### Simulation Round 1: key findings and solutions

The key findings and solutions noted in Round 1 of simulations are summarized in Table 2. For event recording, we identified inaccurate recording as the primary problem and determined more training and practice would resolve this. Other usability issues, such as difficulty with a pop-up window, clicking on checkboxes, and using toggle buttons, were also identified; we resolved these issues with updating the user interface (e.g., button enlargement) along with additional training and practice for users.

For real-time guidance, we noted that participants did not respond to two key audio cues: "suction only if necessary," triggered when suctioning was excessive, and "take corrective steps," triggered when ventilation was not effective (Table 2). To address this, we made multiple modifications. Guidance for the suction cascade was refined to be more definitive, saying "stop suctioning when secretions are cleared." Text for improving ventilation was edited to say, "reposition the head and replace the mask." Trigger criteria on the app to provide guidance for improving ventilation was altered so that it would not be given when heart rate was increasing but below 100.

We also added advice during effective ventilation to 'monitor breathing while ventilating.' As it was observed that visual feedback was rarely used, including the clock displayed by Liveborn, we added audio prompts for time elapsed since birth. During the stillbirth scenario, we noted the guidance felt inappropriate once the midwives had determined the case was a stillbirth; we addressed this in training by proposing midwives discontinue use of the app after stillbirth is clinically confirmed.

For debriefing, the flow of the screens and the questions for discussion were irrelevant for stillbirth cases as they did not account for the influence of a clinical diagnosis of stillbirth on the provider's actions. In response, we developed a new flow for debriefing that considers the outcome of the case at the outset with debriefing screens tailored for this outcome (Fig 2). Additionally, we noted that providers had difficulty navigating to the debriefing screen; to

**Table 1. Demographics of participants.**

| Demographics | Midwives N=10 |
|---|---|
| Age, median (Q1, Q3) | 55.5 (48.3, 61.0) |
| Training | |
| Midwife | 30% |
| Registered nurse | 70% |
| Highest level of education | |
| Baccalaureate | 50% |
| Secondary school | 10% |
| License | 40% |
| >10yrs clinical experience | 100% |
| Type of device owned | |
| Smartphone | 50% |
| Basic cellphone | 50% |
| Laptop computer | 0% |
| Tablet computer | 0% |
| E-book reader | 0% |
| Fitness tracker or smart watch | 0% |
| Other | 0% |
| Frequency of app usage among smartphone owners (n=5) | |
| More than once a day | 0% |
| About once a day | 0% |
| A few times per week | 80% |
| About once per week | 0% |
| Less than once per week | 0% |
| Do not use | 20% |
| Frequency of app or internet usage for work among smartphone owners (n=5) | |
| More than once a day | 0% |
| About once a day | 0% |
| A few times per week | 60% |
| About once per week | 0% |
| Less than once per week | 0% |
| Do not use | 40% |

facilitate easier navigation, we added navigation to debriefing on the history screen of the app (Table 2).

## Simulation Round 2: key findings and solutions

The key findings and solutions noted in Round 2 are summarized in Table 3. We discovered inaccuracies in recording improved but remained a problem. We determined this would be best addressed with ongoing training and practice among providers.

For real-time guidance, responses to the suction cascade and the guidance to take corrective steps during ventilation were improved (Table 3). We noted that advice to 'continue ventilation' was repetitive once BMV began when it would be more clinically appropriate to prompt corrective steps; as such, we altered the trigger criteria to intermittently prioritize guidance to take corrective steps over guidance to continue BMV.

**Table 2. Simulation Round 1: key findings and solutions.**

|  | Key Finding | Solution |
|---|---|---|
| **Event recording** | | |
| **Accuracy** | Inaccurate recording of provider actions*† | Training and practice |
| **Breathing status** | Breathing button rarely used*†‡ | Fabrication of simulation; study during clinical pilot |
| **Distraction with RT feedback** | None* | n/a |
| **Ending case** | Difficulty with pop-up box that ends a case* | Training and practice |
| **Summary screen** | Difficulty clicking check boxes/toggle buttons* | Enlarge |
|  | Confusion re: question about number of providers*† | Add info button |
|  | Required prompting for most questions on the screen* | Change flow to feel more 'clinical' than 'research' |
| **RT Guidance** | | |
| **Response to guidance** | Did not respond to 'suction only if necessary'*† | Alter suction cascade text to be more directive |
|  | During BMV, did not respond to 'take corrective steps'*† | Alter BMV cascade text to make more specific |
| **Appropriateness of guidance** | Guidance felt inappropriate for SB scenario*†‡§ | Instruct users that guidance will never state to stop resuscitating; guidance should be discontinued once SB is clinically confirmed |
|  | Ineffective BMV advice given when HR <100 bpm but increasing*† | Alter trigger criteria for ineffective BMV to include increasing HR |
|  | Guidance that "Ventilation is not effective; call for help" is unclear regarding whether BMV should be continued | Change guidance to "Continue ventilation. Call for help." |
| **Missing guidance** | Long pause with no advice during effective BMV*† | Add 'monitor breathing while ventilating' |
| **Use of visual feedback** | Rarely used*§ | Add audio prompts for time elapsed |
| **Distraction with RT feedback** | None* | n/a |
| **Debriefing** | | |
| **Navigation** | Not readily apparent how to access* | Add to history screen |
| **Debriefing discussion** | Irrelevant questions for SB*† | New flow based on outcome |
|  | Question re: number of providers confusing*† | Re-word |
|  | Did not record anything in text box*† | Emphasize in second round of simulations |
| **Interaction between provider and observer** | Observer drove screens* | Have provider hold tablet for second round |

Abbreviations: BMV=bag-mask ventilation; HR=heart rate; RT=real-time; SB=stillbirth

* Noted from audio video data; † Noted from Liveborn app data; ‡ Noted from observer surveys; § Noted from provider surveys

For debriefing, the new debriefing flow based on outcome improved relevance of the questions for stillbirth scenarios. We noted difficulty reading the screens out loud; per the participant's request, we refined the app to include an audio option. We also updated the user interface with the possibility to view the entire resuscitation timeline (e.g., adding a scroll bar). To prevent the ability to inadvertently change the outcome on the introductory screen, we added a question to confirm when outcome is changed.

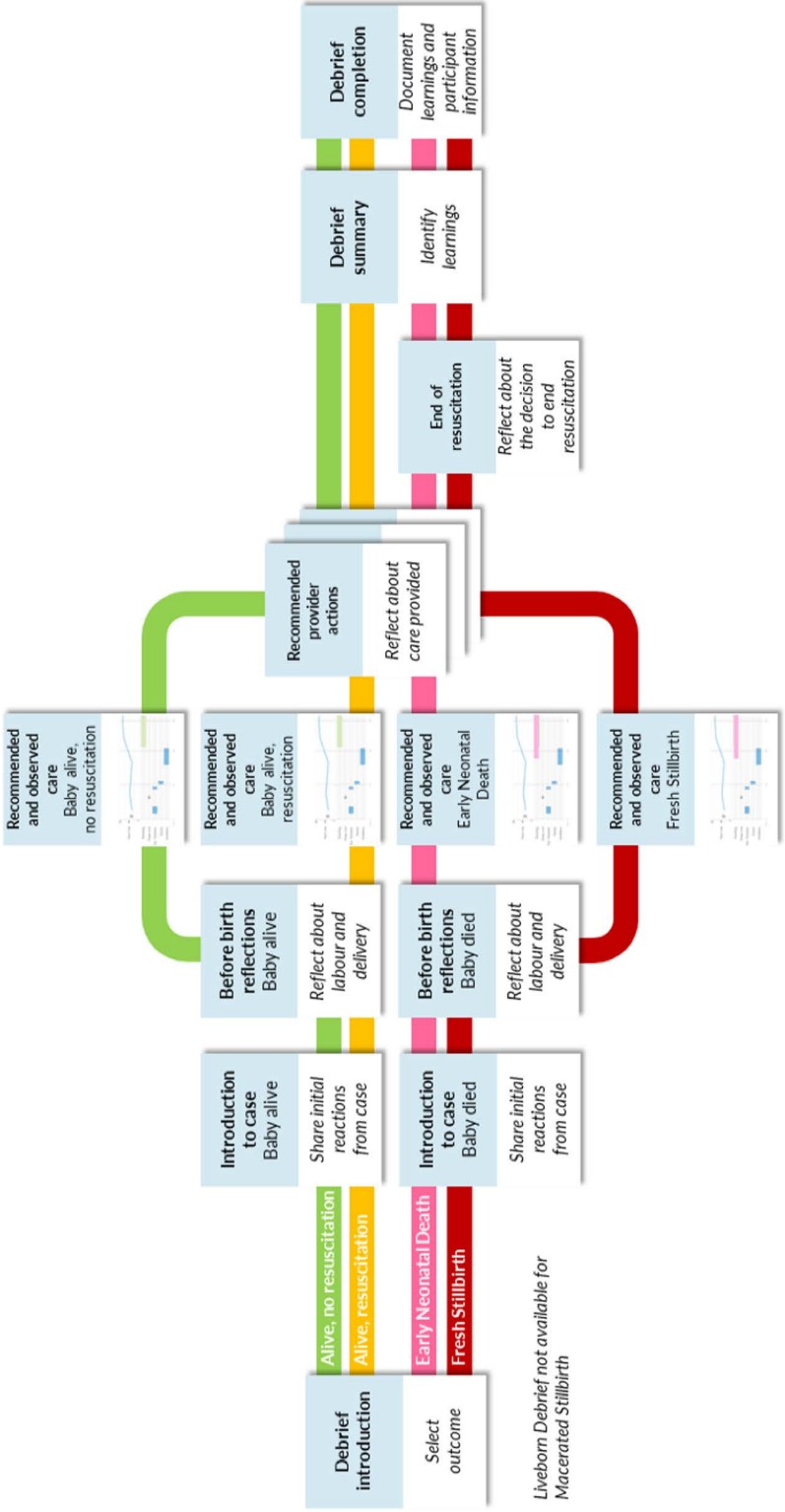

**Fig 2. Debrief pathways based on outcome and observed care.** Before initiating debriefing, the provider enters the outcome of the case (alive, early neonatal death, fresh stillbirth, macerated stillbirth). Based on this outcome and the observed care, the Liveborn app facilitates debriefing through a series of screens that prompt the provider to share initial reactions, review what happened during the case, compare the care provided with Helping Babies Breathe recommendations, and identify learnings and opportunities for improvement. Debriefing is not available for macerated stillbirth.

**Table 3. Simulation Round 2: key findings and solutions.**

| | Key Finding | Solution |
|---|---|---|
| **Event recording** | | |
| **Accuracy** | Inaccurate recording* | Training and practice |
| | Forgot to start observation, pressed dry/stimulation button despite being greyed out* | Make event recording buttons even more faded |
| **Breathing status** | Breathing button rarely used*†‡ | Fabrication of simulation; study during clinical pilot |
| **Summary screen** | Confusion re: question about number of providers*† | Delete question |
| **RT Guidance** | | |
| **Response to guidance** | Response to suction cascade improved*† | n/a |
| | Response to BMV corrective steps improved*† | n/a |
| **Appropriateness of guidance** | Guidance felt inappropriate for SB scenario*†‡§ | Instruct users that guidance will never state to stop resuscitating; guidance should be discontinued once SB is clinically confirmed |
| | Repetition of 'continue ventilation' when more appropriate to prompt corrective steps*† | Return to corrective steps intermittently |
| **Missing guidance** | None*† | n/a |
| **Distraction with RT feedback** | Assisted observer to correct erroneous observations* | n/a |
| **Debriefing** | | |
| **Navigation** | Required prompting to access* | Training |
| **Debriefing discussion** | Questions for SB now relevant*† | n/a |
| | Could not see entire timeline* | Add scroll bar |
| | Inadvertently changed selection of outcome on intro screen* | Add confirmation screen |
| | Question re: number of providers confusing*† | Delete question |
| | Challenge reading the screens* | Add audio |
| **Interaction between provider and observer** | Observer drove screens* | This will be more natural clinically given the time that will elapse between observations and debriefing |

Abbreviations: BMV=bag-mask ventilation; RT=real-time; SB=stillbirth

* Noted from audio video data; † Noted from Liveborn app data; ‡ Noted from observer surveys; § Noted from provider surveys

## Usability and feasibility

The median usability score measured by the SUS was 81 (77, 86) in Round 1 and 90 (85, 95) in Round 2 (p=0.23; Fig 3). The median feasibility score measured by the FIM was 18 (18, 19) in Round 1 and 19 (16, 20) in Round 2 (p=0.94; Fig 4).

## Discussion

Through an iterative process of simulated resuscitations with midwives, we identified usability issues and refined the Liveborn app. The refined app had sufficient usability and feasibility to proceed to a clinical pilot.

This usability evaluation of real-time guidance highlighted the need to focus on language and specificity of the guidance. During a clinical resuscitation, although Liveborn collects

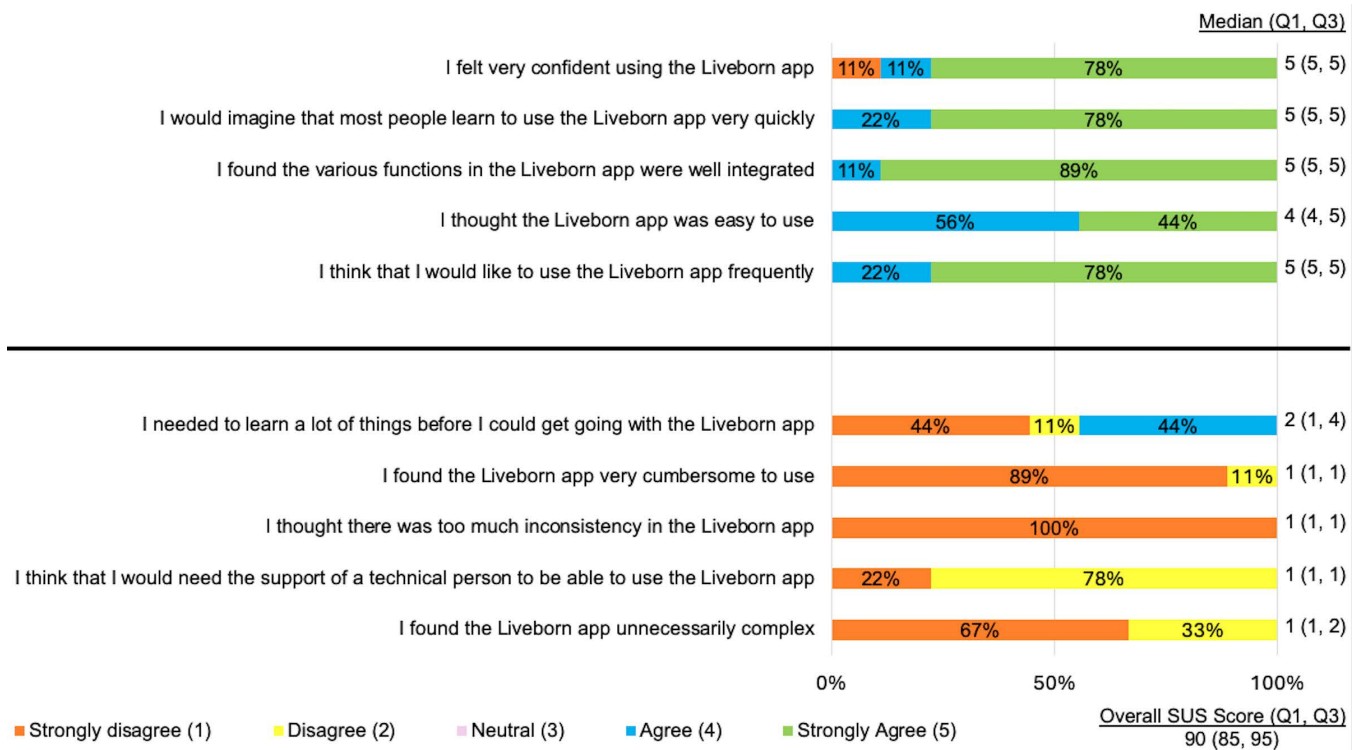

**Fig 3. Usability of Liveborn Feedback.** Usability scores for midwives (n=9 surveyed) following Round 2 of simulated neonatal resuscitations. An SUS score >80 is considered excellent, 68-80 is considered good, 68-58 is considered okay, 51-58 is considered poor and less than 51 is considered awful.

key data elements, we recognize that there are additional nuanced information that could influence clinical decision-making (e.g., the newborn's color and tone). In our initial version of Liveborn, we attempted to recognize this limitation by crafting language for real-time guidance which was supportive of the underlying clinical expertise of the user. When this type of language was tested in simulations, we noted inconsistent adherence to the guidance. For example, when our intent was to prompt discontinuation of suctioning with the phrase "suction only if necessary," midwives continued to suction. When we altered the suction cascade to use the more directive language of "stop suctioning once secretions are cleared," we saw improved adherence. Similarly, when guidance encouraged providers to take corrective steps for ventilation, we found they did not take corrective steps. More specific guidance to reposition the head and mask resulted in improved adherence. The method of delivery of guidance—including its tone, urgency and phrasing—requires further consideration [24].

Compared to real-time guidance, debriefing requires more active interaction with the app on the part of the user. As such, we observed more challenges related to the user interface with debriefing than real-time guidance. Some challenges may have been influenced by infrequent experience with smartphone applications, potentially leading to difficulty navigating the debriefing screens and difficulty accurately recording events. A similar study of the observational component of the Liveborn app in Nepal noted that a provider's familiarity with the use of smartphone applications facilitated use of the app [25]. Of particular note in our evaluation was the midwives' suggestion to enhance Liveborn with an audio option for debriefing. While French is the primary language of the healthcare professional in the DRC and a French version of Liveborn was used, many are more comfortable speaking in Lingala. This may have contributed to midwives finding it burdensome to read the screens in French.

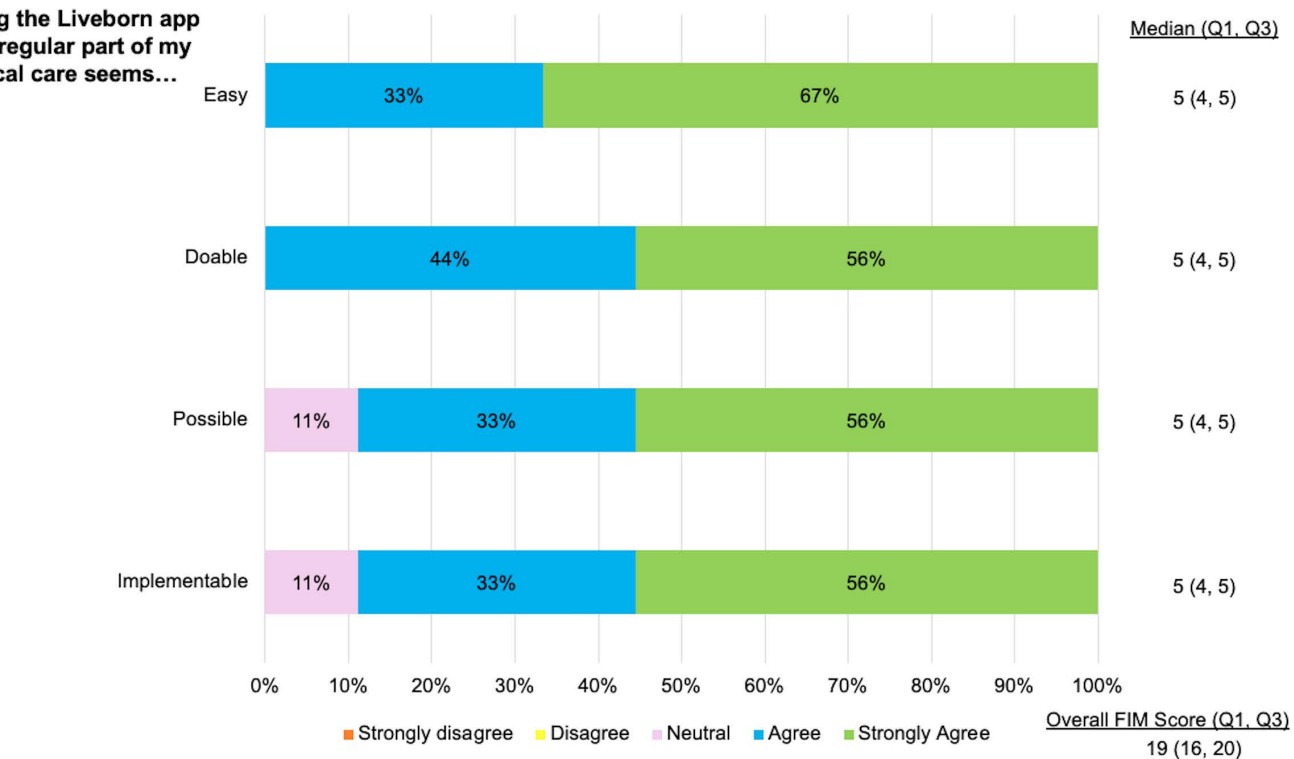

**Fig 4. Feasibility of Liveborn Feedback.** Feasibility of intervention measure scores for midwives (n=9 surveyed) following Round 2 of simulated neonatal resuscitations. 1=strongly disagree, 2=somewhat disagree, 3=neutral, 4=somewhat agree, 5=completely agree.

Our simulations uncovered significant opportunities to refine both real-time guidance and debriefing for stillborn cases. For example, HBB recommends initiating resuscitation for all suspected intrapartum stillbirth infants, advising providers to evaluate for signs of life later in the resuscitation if no clinical response to resuscitation is seen. In simulation, we noted advice provided by real-time guidance felt inappropriate for stillbirth cases. As a decision that care is futile should rest with the clinician and not the tool, we determined this issue should be addressed with training rather than a change in the guidance. As such, we instructed midwives to discontinue use of the guidance if they considered care to be futile, including in cases of stillbirth. Similarly, we noted the language of the debriefing screens presumed a liveborn infant and thus seemed inappropriate for stillborn cases. We refined the language for all debriefing cases by soliciting the outcome of the resuscitation at the outset and developing different debriefing pathways based on this outcome.

Midwives assessed the refined Liveborn app to have excellent usability per the median score of 90 on the SUS. Midwives also assessed the refined Liveborn app to be highly feasible to incorporate into their clinical care with a median score of 19 on the FIM.

Our methodology for evaluating the usability of an mHealth application that provides clinical decision support should be considered in the evaluation of other mHealth tools. We designed simulations to address common clinical errors and test a variety of app features which allowed us to thoroughly evaluate the application and its many facets. We used multiple data streams and triangulation of the data to identify problems with the app. While many of our key findings were identified from video analysis, provider insights via surveys were important in further characterizing identified problems and suggesting solutions. The iterative nature of our testing with app refinement between rounds allowed us to develop a usable tool that can now be tested in a clinical setting.

At the same time, we also note a few limitations. First, there may be questions about generalizability. Our evaluations were derived from a relatively small number of participants—with limited smartphone experience—from a single health facility in Kinshasa. We felt this approach was reasonable in this early evaluation stage but, if its use were expanded, we anticipate further revision and adaptation. Second, our findings related to behavior change during real-time guidance in simulation may not translate to behavior change in a clinical environment; the findings also may have been influenced by the experienced cohort of frontline providers who each had provided clinical care for more than 10 years. Finally, use of the tool in the clinical environment may alter perceptions of its usability and feasibility. As such, a critical next step for evaluation of the tool is testing it clinically.

## Conclusion

The Liveborn mHealth app, after iterative refinement, demonstrated excellent usability and feasibility in simulated neonatal resuscitations. Further research is needed to assess Liveborn's effectiveness in real clinical settings and its impact on neonatal outcomes in LMICs. Following a clinical pilot study of Liveborn, we are now evaluating its effectiveness in a cluster randomized trial in the DRC [26,27].

## Supporting information

**S1 Table. Simulation design targeting common resuscitation errors and a variety of application features.**
(DOCX)

## Author contributions

**Conceptualization:** Daniel Ishoso, Eric Mafuta, Carl Bose, Benjamin H. Chi, Helge Myklebust, Antoinette Tshefu, Jackie K. Patterson.

**Data curation:** Daniel Ishoso, Ingunn Haug, Jackie K. Patterson.

**Formal analysis:** Daniel Ishoso, Abigail McRea, Jackie K. Patterson.

**Funding acquisition:** Jackie K. Patterson.

**Investigation:** Daniel Ishoso, Eric Mafuta.

**Methodology:** Kourtney Bettinger, Ingunn Haug, Patricia Gomez, Joar Eilevstjønn, Jackie K. Patterson.

**Project administration:** Daniel Ishoso.

**Software:** Ingunn Haug, Joar Eilevstjønn.

**Supervision:** Carl Bose, Benjamin H. Chi, Antoinette Tshefu.

**Visualization:** Ingunn Haug.

**Writing – original draft:** Daniel Ishoso, Abigail McRea, Jackie K. Patterson.

**Writing – review & editing:** Daniel Ishoso, Eric Mafuta, Kourtney Bettinger, Carl Bose, Benjamin H. Chi, Ingunn Haug, Patricia Gomez, Joar Eilevstjønn, Abigail McRea, Helge Myklebust, Antoinette Tshefu, Jackie K. Patterson.

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
