## [Decision Letter · Decision Letter 0]

6 Jan 2025

PDIG-D-24-00572Usability Evaluation of the Liveborn App:

a Mobile Health Application that Provides Feedback for Newborn ResuscitationPLOS Digital Health Dear Dr. Patterson, Thank you for submitting your manuscript to PLOS Digital Health. After careful consideration, we feel that it has merit but does not fully meet PLOS Digital Health's publication criteria as it currently stands. Therefore, we invite you to submit a revised version of the manuscript that addresses the points raised during the review process. Please submit your revised manuscript within 60 days Mar 07 2025 11:59PM. If you will need more time than this to complete your revisions, please reply to this message or contact the journal office at digitalhealth@plos.org. Please include the following items when submitting your revised manuscript:* A rebuttal letter that responds to each point raised by the editor and reviewer(s). You should upload this letter as a separate file labeled 'Response to Reviewers '. This file does not need to include responses to any formatting updates and technical items listed in the 'Journal Requirements' section below.* A marked-up copy of your manuscript that highlights changes made to the original version. You should upload this as a separate file labeled 'Revised Manuscript with Track Changes '.* An unmarked version of your revised paper without tracked changes. You should upload this as a separate file labeled 'Manuscript '. If you would like to make changes to your financial disclosure, competing interests statement, or data availability statement, please make these updates within the submission form at the time of resubmission. Guidelines for resubmitting your figure files are available below the reviewer comments at the end of this letter. We look forward to receiving your revised manuscript. Kind regards, Haleh AyatollahiSection EditorPLOS Digital Health Haleh AyatollahiSection EditorPLOS Digital Health Leo Anthony CeliEditor-in-ChiefPLOS Digital Healthorcid.org/0000-0001-6712-6626 **Additional Editor Comments (if provided):****Reviewers' Comments:** Reviewer's Responses to Questions

**Comments to the Author**

1. Does this manuscript meet PLOS Digital Health’s publication criteria ? Is the manuscript technically sound, and do the data support the conclusions? The manuscript must describe methodologically and ethically rigorous research with conclusions that are appropriately drawn based on the data presented.

Reviewer #1: Yes

Reviewer #2: Yes

2. Has the statistical analysis been performed appropriately and rigorously?

Reviewer #1: N/A

Reviewer #2: N/A

3. Have the authors made all data underlying the findings in their manuscript fully available (please refer to the Data Availability Statement at the start of the manuscript PDF file)?

Reviewer #1: Yes

Reviewer #2: No

4. Is the manuscript presented in an intelligible fashion and written in standard English?

Reviewer #1: Yes

Reviewer #2: Yes

5. Review Comments to the Author

Reviewer #1: I highly value the development effort on the liveborn app, as this is likely impactful in both LMIC setting as well as elsewhere. I have read this paper with a background on perinatal clinical research, with some, but limited experience on app development in clinical care. The specific value hereby relates to the integration with eg HBB construct.

Post hoc, I perceive the title somewhat misleading, as the focus is rather on preclinical (simulations) product optimization and development, not similar to usability evaluation (as this suggests in my perception clinical use). In my opinion, the usability in real life (efficacy versus effectiveness) will likely have additional issues like manpower – second pair of hands needed, resources, educational level needed etc.

The tool is likely rather ready to assess usability and feasibility in real life ?

Specific:

In your ‘lay summary’, you rightfully use failure to breathe at birth, and this description is in my opinion more accurate than the ‘respiratory depression’ in the abstract.

Were the same or other midwives recruited for the first and second round of the app ?

Why 10 similations (2 x 5) per round. Was this based on any ‘power’, or rather pragmatic ?

Reviewer #2: In their manuscript, Ishoso et al. describe a usability study of the Liveborn app, which was carried out in the Democratic Republic of Congo. The goal of Liveborn is to improve neonatal outcomes by providing real-time feedback and debriefing for neonatal resuscitation in low-resource settings. The overall Liveborn project has the potential to meaningfully improve global health, and I look forward to reading the results from their ongoing cluster randomised trial.

Overall, the manuscript is clear and well-written, and the authors should be commended for readily discussing both the strengths and the limitations of their usability study. The work highlights the importance of iterative design in mHealth applications. I have no major concerns but I do have a number of questions and minor comments which I wish were addressed in the text.

1. Data sharing

There are significant commercial implications of this work, which is presumably why the manuscript does not describe the interface of the Liveborn app in detail. This is understandable, but when the authors state that “all data files will be available …”, what data is this referring to?

2. Languages

The French version of Liveborn is used in this study. What other languages is Liveborn available in? Additionally, how does the team think about and ensure quality of translation?

3. Guidelines used

What are the resuscitation guidelines / clinical algorithms underpinning Liveborn? The authors mention Helping Babies Breathe (L84), and I understand this is based on ILCOR guidelines. However, I can imagine that in different hospitals, slightly different resuscitation algorithms may be used. Can the app be customised to take this into account? How likely is it that the app recommends action that is different from established practice at the site it is used in?

4. Integration with NeoBeat

It is not immediately clear whether NeoBeat is necessary for use of the Liveborn app. Can Liveborn be used without continuous heart rate monitoring of the neonate? Or using another heart rate monitoring device? I imagine that in a low-resource setting, neonatal heart rate monitoring is not always available.

5. Paired work

During both arms of the usability study (real-time feedback and debrief), midwives worked in pairs. Is the intention that Liveborn is used like this in the clinic, and how feasible is it that every time Liveborn is used in the clinic, a qualified midwife or nurse would be available as an observer who records data and interacts with the app?

6. SUS improvement

Is the System Usability Score improvement between rounds 1 and 2 statistically significant?

7. Participant demographics

What was the reason participants were overwhelmingly older and universally very experienced / why were more junior midwives not included in the study? If the goal of Liveborn is to improve adherence to resuscitation guidelines, feedback from more junior clinicians would be essential.

8. Inaccuracies in recording

Even after Round 2, authors found inaccuracies in data recording persisted. They suggest further training could improve this (L234-236), which I am somewhat sceptical of. How common and how problematic is the issue of inaccuracy with the updated app interface?

9. Figure quality

Please not figure resolution is not sufficient for full-page viewing.

6. PLOS authors have the option to publish the peer review history of their article (what does this mean? ). If published, this will include your full peer review and any attached files.

**Do you want your identity to be public for this peer review?** For information about this choice, including consent withdrawal, please see our Privacy Policy .

Reviewer #1: **Yes: ** karel allegaert

Reviewer #2: No

---

## [Decision Letter · Decision Letter 1]

5 Mar 2025

Preclinical Usability Evaluation of the Liveborn App: A Mobile Health Application that Provides Feedback for Neonatal Resuscitation

PDIG-D-24-00572R1

Dear Dr. Patterson,

We are pleased to inform you that your manuscript 'Preclinical Usability Evaluation of the Liveborn App: A Mobile Health Application that Provides Feedback for Neonatal Resuscitation' has been provisionally accepted for publication in PLOS Digital Health.

Best regards,

Haleh Ayatollahi

Section Editor

PLOS Digital Health

**Additional Editor Comments (if provided):**

**Reviewer Comments (if any, and for reference):**

Reviewer's Responses to Questions

**Comments to the Author**

1. If the authors have adequately addressed your comments raised in a previous round of review and you feel that this manuscript is now acceptable for publication, you may indicate that here to bypass the “Comments to the Author” section, enter your conflict of interest statement in the “Confidential to Editor” section, and submit your "Accept" recommendation.

Reviewer #1: All comments have been addressed

Reviewer #2: All comments have been addressed

2. Does this manuscript meet PLOS Digital Health’s publication criteria ? Is the manuscript technically sound, and do the data support the conclusions? The manuscript must describe methodologically and ethically rigorous research with conclusions that are appropriately drawn based on the data presented.

Reviewer #1: Yes

Reviewer #2: Yes

3. Has the statistical analysis been performed appropriately and rigorously?

Reviewer #1: Yes

Reviewer #2: Yes

4. Have the authors made all data underlying the findings in their manuscript fully available (please refer to the Data Availability Statement at the start of the manuscript PDF file)?

Reviewer #1: Yes

Reviewer #2: (No Response)

5. Is the manuscript presented in an intelligible fashion and written in standard English?

Reviewer #1: Yes

Reviewer #2: Yes

6. Review Comments to the Author

Reviewer #1: the suggestions provided have been considered and were addressed

Reviewer #2: (No Response)

7. PLOS authors have the option to publish the peer review history of their article (what does this mean? ). If published, this will include your full peer review and any attached files.

**Do you want your identity to be public for this peer review?** For information about this choice, including consent withdrawal, please see our Privacy Policy .

Reviewer #1: **Yes: ** karel allegaert

Reviewer #2: **Yes: ** Lyuba V. Bozhilova
